# FedFDD: Federated Learning with Frequency Domain Decomposition for Low-Dose CT Denoising

**Xuhang Chen**[1,2]                       XC369@CAM.AC.UK

**Zeju Li**[3]                      ZEJU.LI@NDCN.OX.AC.UK

**Zikun Xu**[1]                  ZIKUN.XU22@IMPERIAL.AC.UK

**Kaijie Xu**[1]                    NORTROM@BERKELEY.EDU

**Cheng Ouyang**[4,5]                C.OUYANG@IMPERIAL.AC.UK

**Chen Qin**[1]                    C.QIN15@IMPERIAL.AC.UK

[1] *Department of Electrical and Electronic Engineering & I-X, Imperial College London, UK*

[2] *Department of Clinical Neurosciences, University of Cambridge, Cambridge, UK*

[3] *Nuffield Department of Clinical Neurosciences, University of Oxford, UK*

[4] *Institute of Clinical Sciences, Imperial College London, UK*

[5] *Department of Engineering Science, University of Oxford, UK*

**Editors:** Accepted for publication at MIDL 2024

## Abstract

Low-dose computed tomography (LDCT) enables imaging with minimal radiation exposure but typically results in noisy outputs. Deep learning algorithms have been emerging as popular tools for denoising LDCT images, where they typically rely on large data sets requiring data from multiple centers. However, LDCT images collected from different centers (clients) can present significant data heterogeneity, and the sharing of them between clients is also constrained by privacy regulations. In this work, we propose a personalized federated learning (FL) approach for enhancing model generalization across different organ images from multiple local clients while preserving data privacy. Empirically, we find that earlier FL methods tend to underperform single-set models on non-IID LDCT data due to the presence of data heterogeneity characterized by varying frequency patterns. To address this, we introduce a Federated Learning with Frequency Domain Decomposition (FedFDD) approach, which decomposes images into different frequency components and then updates high-frequency signals in an FL setting while preserving local low-frequency characteristics. Specifically, we leverage an adaptive frequency mask with discrete cosine transformation for the frequency domain decomposition. The proposed algorithm is evaluated on LDCT datasets of different organs and our experimental results show that FedFDD can surpass state-of-the-art FL methods as well as both localized and centralized models, especially on challenging LDCT denoising cases. Our code is available at https://github.com/xuhang2019/FedFDD.

**Keywords:** Federated Learning, Low-Dose CT Denoising, Discrete Cosine Transform.

## 1. Introduction

Federated learning (FL) is a decentralized approach that can integrate data from multiple local clients for training machine learning models in a privacy-preserving way (Pati et al., 2022). FL is particularly well-suited for medical imaging tasks that require extensive data, where privacy is a major concern. For example, FL has been discussed for applications including COVID-19 classification (Dayan et al., 2021), skin lesion classification (Yan et al., 2023), and human activity recognition (Rieke et al., 2020; Rauniyar et al., 2023).

Machine learning models typically rely on large training data sets for achieving good generalization capability. Low-dose Computed Tomography (LDCT) denoising models have been shown to perform better by expanding the training dataset with samples from different sources and various anatomies (Yang et al., 2023; Immonen et al., 2022). Nevertheless, a centralized training paradigm may not be always feasible, as local clients may be reluctant to share data due to privacy concerns. Hence, in this study, we propose to investigate a novel LDCT denoising method that can leverage data from diverse clients in a privacy-preserving way, with each personalized local model specializing in different anatomical regions.

**Related works.** Fruitful research has been proposed in the LDCT denoising field. RED CNN (Chen et al., 2017) adopts residual skip connection and small convolution kernels to enable smooth variation on different layers of feature maps for LDCT denoising. Transformer-related works have also been proposed to utilize the pretrained data (Jing et al., 2022) and enrich the diversity and effectiveness of features (Wang et al., 2023). Similarly, adversarial learning methods (Wolterink et al., 2017; Han et al., 2022) have also been studied to learn the denoising task. However, few of them addressed the CT denoising issue in the FL settings. Current FL approaches mainly focus on addressing classification or segmentation challenges rather than tasks related to image restoration. FedAvg (McMahan et al., 2017) proposed to iteratively average the model weights from local models following each local update step, assuming that all clients have independent and identically distributed (IID) data, which however is often not the case in real-world scenarios. To mitigate the sub-optimal performance caused by non-IID data, FedProx (Li et al., 2020) integrated a proximal term to the objective function to guide the local updates to align with the global model, thereby improving convergence with heterogeneous data. FedBN (Li et al.) was proposed to freeze the Batch Normalization layer when training the domain-shifted medical data for stabilizing the training and thus improving the averaging model. Besides, MOON (Li et al., 2021) has added a contrastive loss from the latent vector of each client to the global model, which aims to alleviate the global optimal parameter deviation issue by reducing the imbalanced gradient drift.

More recently, FL methods in low-level medical imaging tasks have also raised attention, such as in image reconstruction. For instance, FedMRI (Feng et al., 2023b) has proposed to divide the reconstruction model into a global-shared encoder and separate local-preserved decoders with a weighted contrastive regularization, which has been shown to improve the efficiency and accuracy of MRI reconstruction tasks. FedPR (Feng et al., 2023a) further proposed a federated paradigm to only communicate the pre-trained-model-generated prompts and optimize them in an approximate null space of global prompts. Of particular relevance to our work, HyperFed (Yang et al., 2022) has proposed to utilize localized hypernetworks based on simulated geometric parameters and dose levels to guide the CT reconstruction task, i.e., reconstructing the projection data to the imaging data. In contrast, our work focuses on the LDCT denoising with various anatomies in an FL setting, and the design of our proposed model is inherently inspired by the denoising nature.

**Our contributions.** We propose a Federated Learning with Frequency Domain Decomposition (FedFDD) strategy for training the LDCT denoising model in a privacy-preserving way. FedFDD is motivated by the observation that LDCT images share common patterns across different anatomies, particularly in high-frequency components (*i.e.*, noises). Therefore, we aim to leverage that and enhance the learning of the denoising task in the general

high-frequency domain using an FL approach while preserving individual low-frequency components (*i.e.* semantic anatomical structures) (Yang et al., 2022) for each local client.

Our contributions are mainly threefold: 1) We introduce a novel dual-path FL strategy with frequency domain decomposition to split the feature space for training and maintain gradient stability during model aggregation under non-IID conditions. 2) We propose to leverage data from different anatomies in LDCT denoising tasks, to mitigate data scarcity in FL settings. 3) We demonstrate that selectively updating high-frequency components in a dual setting significantly enhances the model's performance in LDCT denoising tasks, with competitive performance against state-of-the-art FL methods. Our proposed approach aligns with the nature of noise removal and indicates a promising direction in FL for imaging.

## 2. Methods

### 2.1. Federated Learning Problem Formulation

The main goal of our FL method is to utilize datasets from different clients in a privacy-preserving way. Specifically, we aim to construct a model that outperforms localized models, *i.e.,* models trained on local datasets. In order to prevent privacy leakage, we follow the setting that data from different clients cannot be communicated (*i.e.,* different hospitals do not allow patient data transmission). Given that there are $N$ clients with their own datasets $\mathcal{D}^i, i = 1, 2, ..., N$ and the optimization loss function as $\mathcal{L}$, we want to achieve:

$$\underset{\omega_{global}}{\arg\min}(\sum_{k=1}^{N} p_k \mathcal{L}(\mathcal{D}^k; \omega_{global})), \tag{1}$$

where $\omega_{global}$ denotes parameters of a global model, and $p_k$ is the weight of each local dataset, defined as $p_k = ||\mathcal{D}^k|| / \sum_{i=1}^{N} ||\mathcal{D}^k||$ where $||\mathcal{D}^k||$ is the size of the $k$-th dataset.

Previous research (Xu et al., 2023) mentioned that the conventional mini-batch gradient descent method aimed to update the model parameters at $j+1$ time step on the client $k$ by $\omega_k^{j+1} = \omega_k^j - \eta \nabla \mathcal{L}(\mathcal{D}^k; \omega_k^j)$, where $\eta$ is the learning rate and $\nabla$ calculates the gradient w.r.t $\omega_k^j$. In this way, the global model parameters can be aggregated by $\omega_{global}^{j+1} = \sum_{i=1}^{N} p_i \omega_k^{j+1}$. However, the non-IID property of $\mathcal{D}^k$ would generate varying directions of $\nabla \mathcal{L}(\mathcal{D}^k; \omega_k^j)$ and the weighted operation would therefore drift the global parameter to a sub-optimal solution (McMahan et al., 2017).

In our LDCT denoising setting with images of various anatomies from different clients, we observed that the mainstream FL scheme suffered from the drifted direction caused by data of *different anatomical regions*. Therefore, we aim to alleviate the misleading impact of the varying anatomical structures on the denoising task and guide the model to learn the denoising essence. To achieve this, we propose to decompose the denoising task into two paths: one path consists of part of the model that updates the local parameters $\omega_{\text{anatomy}}$ corresponding to refining each anatomical structure, and the other specifically deals with the noise reduction part with model weights of $\omega_{\text{denoise}}$. In this way, the global model will only be updated with the gradient direction of the denoising part $\nabla \mathcal{L}(\mathcal{D}^k; \omega_{k,\text{denoise}}^j)$, without the negative influence from the aggregation of the drifted item $\nabla \mathcal{L}(\mathcal{D}^k; \omega_{k,\text{anatomy}}^j)$. The model parameter at the client $k$ can then be viewed as $\omega_k = \omega_{k,\text{anatomy}} \cup \omega_{k,\text{denoise}}$.

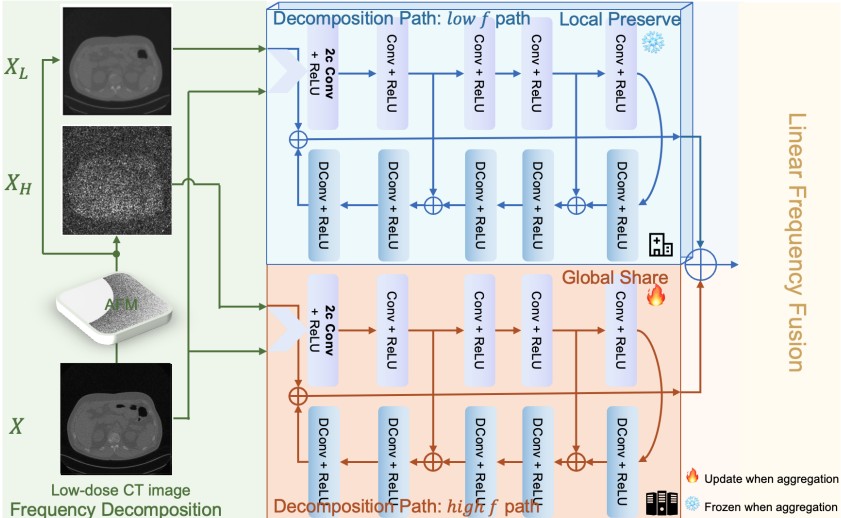

Figure 1: The conceptual diagram of the proposed personalized federated learning models with two frequency fusion methods. The LDCT images are split into different frequency components and the model will update high-frequency signals in an FL setting (the network branch colored in orange) while preserving local low-frequency characteristics (the network branch colored in blue). Both frequency components are then fused to form the output.

Therefore, the optimization problem can be formulated as:

$$\underset{\omega_1,...,\omega_k}{\arg\min}(\sum_{k=1}^{N} p_k \mathcal{L}(\mathcal{D}^k; \omega_k)). \tag{2}$$

During the aggregation at time step $j+1$, $\omega_{\text{denoise}}^{j+1}$ and $\omega_k^{j+1}$ are updated as $\omega_{\text{denoise}}^{j+1} = \sum_{i=1}^{N} p_i \omega_{k,\text{denoise}}^{j+1}$ and $\omega_k^{j+1} = \omega_{\text{denoise}}^{j+1} \cup \omega_{k,\text{anatomy}}^{j+1}$.

## 2.2. Federated Learning with Frequency Domain Decomposition

Based on the above formulation, we propose a dual-path FL strategy for LDCT denoising, as shown in Fig. 1. Noise in LDCT can be represented using various noise models including quantum noise from X-ray (Yang et al., 2023), normally distributed stochastic process noise (Li et al., 2023), and speckle and streak (Yang et al., 2020) noise, which are commonly corresponding to the high-frequency component in the images. On the other hand, the anatomical structures can be represented by the low-frequency component of the images. Motivated by this, we propose to decompose the effects of anatomical structures and noises through frequency domain decomposition, thereby updating a global denoising model component with high-frequency data across different anatomies while preserving personalized local low-frequency model components for each client, as seen in Fig. 1.

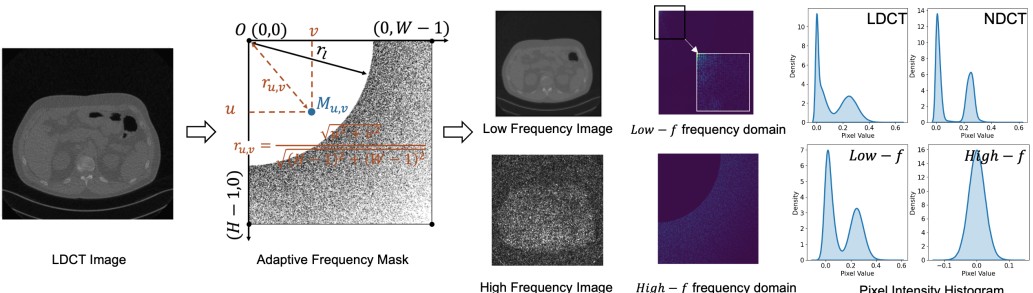

Figure 2: Left: The image and frequency space of LDCT image and its two frequency components. Right: The intensity histogram of different images.

### 2.2.1. FREQUENCY DECOMPOSITION WITH ADAPTIVE FREQUENCY MASK

In detail, for the frequency domain decomposition, we propose to design a mask to decompose the image and utilize an FL strategy to learn the model parameters for the denoising part, $i.e.,$ $\omega_{\text{denoise}}$. Specifically, the Discrete Cosine Transformation (DCT) is adopted for the frequency decomposition due to its real-value property (Fig. 2). The majority of the informative part of the image can then be efficiently condensed into a small number of coefficients with the DCT, especially in the low-frequency coefficients, distributed in the upper-left corner of the DCT frequency domain. The low-frequency part mainly includes semantic-related components of the images ($i.e.,$ the structure of anatomies), and the noise is mainly distributed into the remainder area (Fig. 2).

Therefore, we use a Bernoulli binary mask to separate the semantic component (the low-frequency component) from the noisy area (the high-frequency component) inspired by (Yue et al., 2021). This is achieved by setting a low-frequency threshold $r_l$, within which coefficients are preserved (mask value 1), while those beyond it are determined by a Bernoulli distribution based on their normalized Euclidean distance $r_{u,v}$ to retain potentially useful high-frequency information. The mask can be formulated as:

$$M_{u,v} = \begin{cases} 1, & 0 \le r_{u,v} < r_l \\ \text{Bernoulli}(r_{u,v}), & r_l \le r_{u,v} \le 1 \end{cases}. \tag{3}$$

With the defined binary adaptive frequency mask (Appendix A) $M$ and an input $X$, the low frequency image part $X_L$ and high frequency part $X_H$ can be derived as: $X_L = \mathcal{F}^{-1}(\mathcal{F}(X) \odot M)$, $X_H = \mathcal{F}^{-1}(\mathcal{F}(X) \odot (1-M))$. As both the DCT and the inverse DCT process are linear, we can have $X = X_L + X_H$, which allows for the linear fusion strategy.

### 2.2.2. THE OVERALL MODEL ARCHITECTURE

Fig. 2 demonstrates that the low-frequency components pixel intensity distribution (Low-f) of LDCT correlates well with the distribution of the NDCT image ($i.e.,$ clear, full-dose imaging), while high-frequency components (High-f) predominantly exhibit noise with an IID distribution. Motivated by this, we propose to tackle the non-IID distribution characteristics of the problem by decomposing the model into dual paths, where one network branch (parameterized by $\omega_{k,anatomy}$) is designed to preserve the anatomy locally and the

other branch aims to tackle the noise part using FL, leveraging its effectiveness in dealing with IID data (*i.e.,* high-frequency components in this case).

An illustration of the FedFDD model is shown in Fig. 1. Both the low $f$ and high $f$ backbone models are inherited from the RED-CNN model (Chen et al., 2017). The first convolutional layer outputs a single-channel feature map taking both the frequency component and the original LDCT image as input. This layer can be viewed as a feature fusion layer. The frequency component, exclusively treated as a residual item added to the output of the branch, is expected to drive the branch to learn the intrinsic features within each branch (*i.e.,* anatomies for low frequency and denoising for high frequency). During the aggregation, the low-frequency path (the blue branch) is reserved for each local client, and the high-frequency path (the orange branch) is aggregated. The objective is defined by the Mean Squared Error (MSE) between the reconstructed image and the NDCT standard. The model is trained in an end-to-end way.

## 3. Experiments

### 3.1. Datasets and implementation details

The LDCT and NDCT datasets (Moen et al., 2021) are licensed by The Cancer Image Archive (TCIA) team. Original data is collected by the Mayo Clinic, containing the abdomen, chest, and head regions. We simulated 36 patients with different anatomies as three clients (*Client 1: 12 Abdomen, Client 2: 12 Chest and Client 3: 12 head*). We provide detailed data settings in Appendix B. We divide those datasets into 60% (training), 10% (validation), and 30% (testing). The modified dual-path network is trained with an MSE loss with Adam optimizer for a total of 200 epochs. The learning rate is initialized as $10^{-4}$ and decayed per 3000 iterations. We stop the training process early if the validation loss fails to decrease for 10 consecutive epochs (Qian et al., 2024). The $r_l$ of the mask is 0.45. The Hounsfield Units (HU) window of the CT images is $[-160, 240]$ and the images are normalized to $[0, 1]$ by the minimum of $-1024$ and a maximum of 3072 (Yang et al., 2022; Bera and Biswas, 2023). Patch training is adopted with patch size $64 \times 64$ and a total of 16 per image following (Bera and Biswas, 2021).

### 3.2. Experimental Results and Discussion

#### 3.2.1. Comparison Study

**Limitation of current FL approaches.** Contrary to expectations, state-of-the-art FL methods (e.g., FedAvg, FedProx, MOON, and FedBN) do not surpass localized or centralized training in LDCT denoising tasks, as evidenced by Table 1. Typically, FL is expected to excel by leveraging diverse client data, but the non-IID nature of the LDCT dataset and notable domain shift (Fig. 2) impede this advantage. Current FL approaches mostly do not address these specific challenges and thereby exhibit constrained performance in such scenarios. In contrast, centralized training benefits from comprehensive data exposure, facilitating superior generalization, whereas the effectiveness of localized training is curtailed by its limited dataset scope. This discrepancy is particularly marked in chest dataset comparisons, underscoring the limitations of existing FL approaches in handling non-IID distributions and domain variability.

**Comparison results.** We compare our proposed FedFDD with state-of-the-art FL algorithms in Table 1. Our method outperforms other methods in terms of PSNR and SSIM in

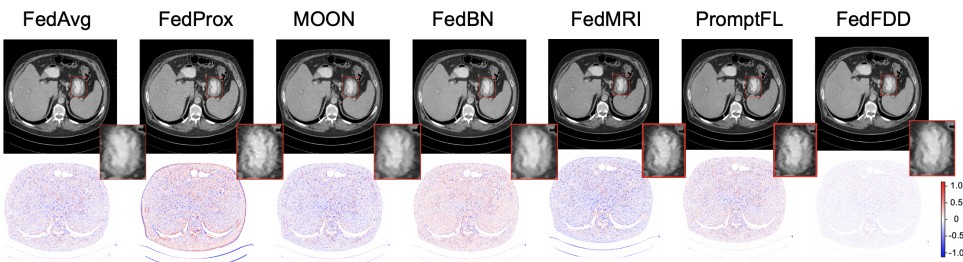

Figure 3: Top: Qualitative examples from different federated algorithms. Bottom: The error maps when compared with the ground truth.

Table 1: Quantitative results of the different methods. The best results are in **bold**.

| Method | Chest | | Abdomen | | Head | |
|---|---|---|---|---|---|---|
| | **PSNR↑** | **SSIM↑** | **PSNR↑** | **SSIM↑** | **PSNR↑** | **SSIM↑** |
| LDCT | 15.3388 | 0.6857 | 28.3165 | 0.8213 | 43.8707 | 0.9652 |
| Localized | 22.1312 | 0.7443 | 32.3983 | 0.8790 | **44.3414** | 0.9756 |
| Centralized | 22.4649 | 0.7534 | 32.4423 | 0.8809 | 43.6559 | **0.9816** |
| FedAvg | 21.3752 | 0.7538 | 31.8950 | 0.8735 | 43.1911 | 0.9788 |
| FedProx | 19.4455 | 0.7087 | 28.8216 | 0.8448 | 35.4972 | 0.9638 |
| MOON | 21.2922 | 0.7509 | 31.4486 | 0.8641 | 40.6419 | 0.9746 |
| FedBN | 21.8685 | 0.7472 | 31.8466 | 0.8716 | 42.2921 | 0.9786 |
| FedMRI [1] | 22.1496 | 0.7452 | 32.2135 | 0.8792 | 42.5368 | 0.9788 |
| PromptFL [2] | 21.9730 | 0.7483 | 31.9425 | 0.8715 | 42.2195 | 0.9782 |
| FedFDD | **22.6209** | **0.7586** | **32.4510** | **0.8823** | 43.0421 | 0.9792 |

most cases, particularly in the more challenging case of Chest CT denoising (McCollough, 2016). Notably, FedFDD brings improvements of up to 7.2 dB PSNR when compared with original LDCT images and 1.2 dB compared with FedAvg on Chest data. Our experimental results indicate the effectiveness of the proposed method, which specifically considers the inherent challenges of LDCT denoising. Undesirable performance in the Head dataset could be attributed to the subtle discrepancy between LDCT and NDCT (Appendix B) where all FL approaches do not outperform the localized training. Despite that, our approach can achieve a higher SSIM compared to baseline methods. In contrast, other advanced FL algorithms underperform the localized training on all three datasets. Furthermore, we visualize the denoised images and their corresponding error maps in Fig. 3. We find that the error produced by FedFDD is significantly less than that of others and the denoised image has well maintained the structural details and textures.

---

1. FedMRI (Feng et al., 2023b) is initially not designed for LDCT denoising. We adapted it to our task for further substantiation. The details can refer to Appendix D.
2. We adapt the idea of HyperFed (Yang et al., 2022) and FedPR (Feng et al., 2023a) to our task, named as PromptFL. We utilize client-specific text embedding to guide the denoising process, as detailed in Appendix D.

| Method | PSNR↑ | SSIM↑ |
|--------|-------|-------|
| **FedFDD**$_{hf}$ | **22.6209** | **0.7586** |
| FedFDD$_{all}$ | 22.3798 | 0.7550 |
| FedFDD$_{lf}$ | 22.3580 | 0.7559 |

Figure 4: Ablation study. Left: Results of different aggregation strategies in frequency-division methods. Right: Sensitivity analysis of the threshold of the adaptive frequency mask $r_l$. Results are shown on Chest data.

### 3.2.2. ABLATION STUDY

**Effects of FL strategy in different frequencies.** Fig. 4 presents the experiments ascertaining the benefits of exclusively updating the high-frequency components. Here, FedFDD$_{all}$ represents a learning strategy that updates both the parameters of the dual-path model during aggregation and FedFDD$_{lf}$ means that we only update the low-frequency path with the high-frequency path frozen during aggregation. Our proposed approach, i.e., FedFDD$_{hf}$, which only updates the high-frequency components, achieves the best performance across all metrics. This indicates that our strategy of focusing on high-frequency updates and freezing the low-frequency components is effective. In contrast, FedFDD$_{lf}$ approach shows a slight deterioration in performance compared with FedFDD$_{all}$. This is likely due to the reason that the merge of low-frequency components from different clients could introduce inconsistencies, leading to a potential loss of specific structural details and thereby affecting the overall image quality. The better performance of FedFDD$_{hf}$ compared with FedFDD$_{all}$ also indicates that the proposed targeted update strategy, focusing on high-frequency components, can be more effective than a holistic update.

**Effects of varying frequency split thresholds.** We show the model performance with varied thresholds $r_l$'s in Fig. 4. Recall that higher $r_l$ refers to less information in the high frequency images. When the threshold is low, the low-frequency component (intrinsic anatomical area) leaks to high frequency updating procedure, which results in a deterioration of the model performance. Besides, there is a slight peak around 0.4 to 0.5, suggesting the optimal value within the range. When $r_l > 0.6$, the performance slightly decreases because the model would benefit less from FL.

### 4. Conclusions

In this study, we proposed the FedFDD model and demonstrated its effectiveness on the LDCT denoising task across different anatomical images. We proposed an FL strategy with frequency domain decomposition, where only the high-frequency components of the network are updated during the aggregation process. This ensures that the intrinsic characteristics of the low-frequency components are preserved locally. Our method achieves up to 1.2 dB improvement compared with the state-of-the-art FL algorithms. Notably, it also outperforms both single-set and centralized training, particularly in more noisy scenarios (e.g., on Chest data). In the future, we plan to validate FedFDD in an out-of-federation setting for data from unseen clients.

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

## Appendix A. Mask Design

Assuming the image size is $H \times W$, the normalized Euclidean distance of a pixel $(u, v)$ can be:

$$r_{u,v} = \frac{\sqrt{u^2 + v^2}}{\sqrt{(H-1)^2 + (W-1)^2}}$$

.

$M_{u,v}$ represents the value at the point $(u, v)$ in the binary mask (*i.e.*, $M_{u,v} \in \{0, 1\}$).

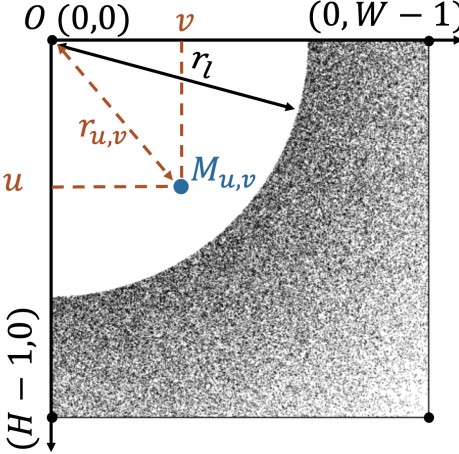

Figure 5: A detailed illustration of the mask

## Appendix B. Detailed Data Information

To ensure the performance, we randomly selected 12 patients in each anatomy. The patient number below indicates the identifiers of patients in the database. Each client data has CT images from both SOMATOM Definition AS+ and SOMATOM Definition Flash scanners.

Client 1: Selected patient number: L143, C004, C012, C027, C030, C050, C067, C002, C016, C021, C052, L506. Each patient has approximately 200 images on average.

Client 2 selected patient number: L067, L096, L192, L286, L310, L033, L049, L056, L109, L291, L014, L019. Each patient has approximately 320 images on average.

Client 3 selected patient number: N012, N024, N030, N047, N051, N053, N072, N076, N079, N082, N085, N100. Each patient has approximately 38 images on average.

An example of the data is:

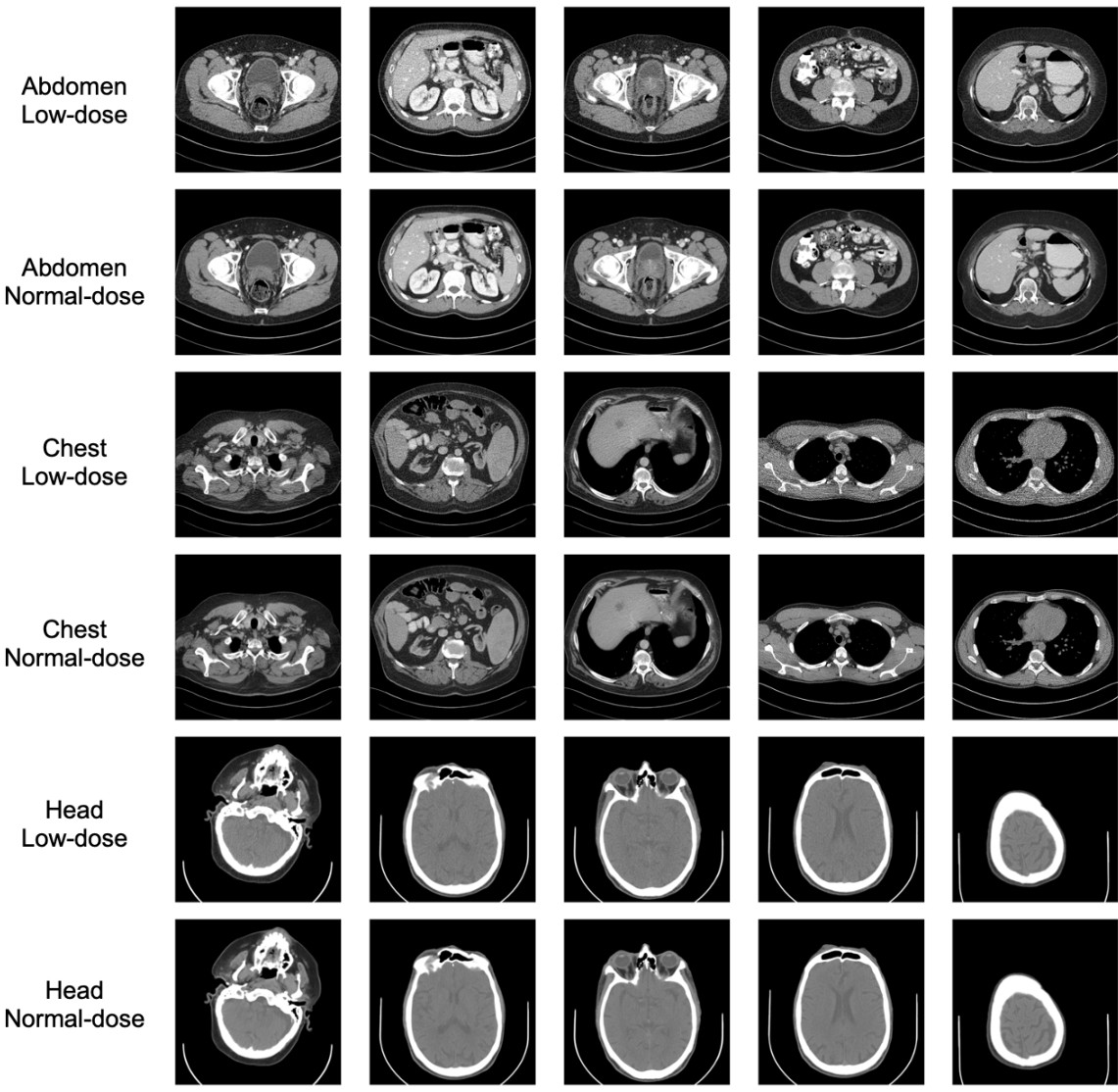

Figure 6: Samples in the dataset from three body parts (clients).

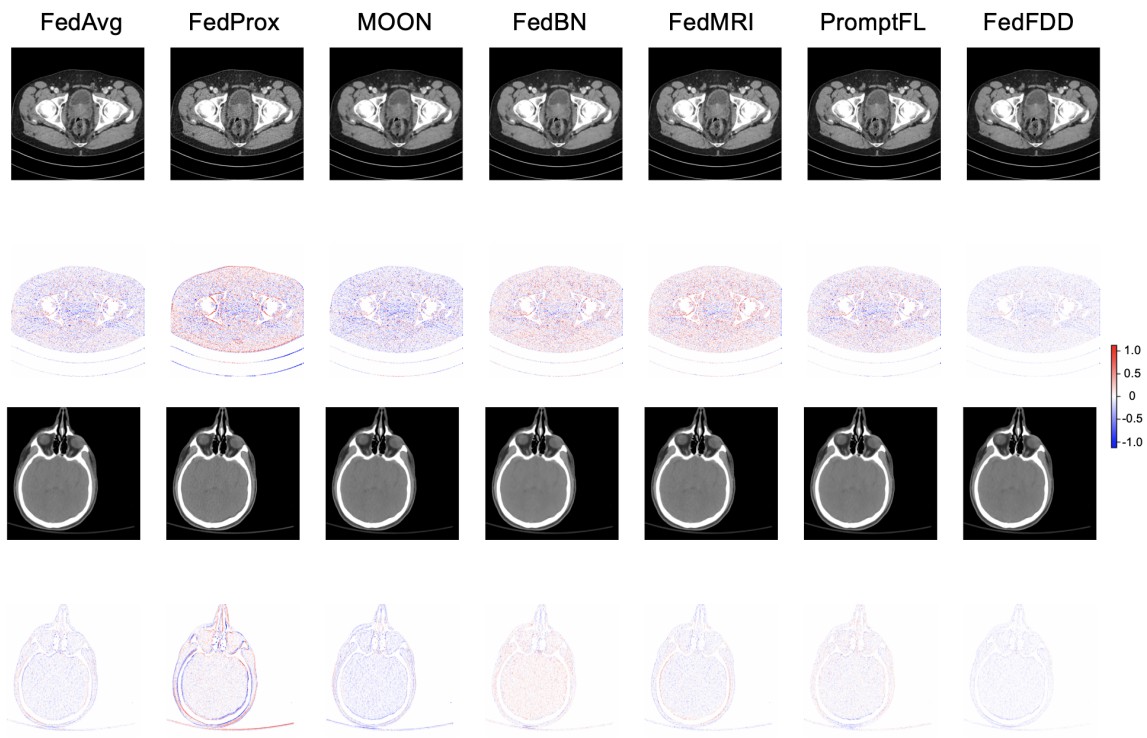

Figure 7: The denoising results of the state-of-the-art FL algorithms with their error map on Abdomen and Head.

## Appendix C. Generalizability Study

To validate our approach's generalizability to FL scenarios where each client may contain data with multiple anatomies, we further perform a client generalization test. Specifically, we propose the training set setting as below, where client 1 contains data from both the chest and abdomen, and client 2 and client 3 contain data from the abdomen and head respectively. It is important to mention that we maintain the test set to be consistent with the original data setting. Consequently, the results are directly comparable to those in Table 1.

- Client 1: Chest + Abdomen 1 (Patient's number: L219, L014, L019)

- Client 2: Abdomen 2 (Patient's number: L067, L096, L192, L286, L310, L033, L049, L056, L109, )

- Client 3: Head

We compare our proposed method against three representative FL approaches (as seen in Table 1) with the new data setting, with the results shown in Table 2. It is observed that in the new data setting where the client contains data with multiple anatomies, our proposed approach FedFDD can still achieve an overall better performance compared to other competing methods. This indicates the good generalizability of our method in alternative data scenarios.

Table 2: Client generalization comparison experiment in the new data setting.

| Method | Chest | | Abdomen | | Head | |
|---|---|---|---|---|---|---|
| | **PSNR↑** | **SSIM↑** | **PSNR↑** | **SSIM↑** | **PSNR↑** | **SSIM↑** |
| LDCT | 15.3388 | 0.6857 | 28.3165 | 0.8213 | 43.8707 | 0.9652 |
| Localized | 22.0033 | 0.7445 | 32.1983 | 0.8776 | **44.3414** | 0.9756 |
| Centralized | 22.4649 | 0.7534 | 32.4423 | 0.8809 | 43.6559 | **0.9816** |
| FedAvg | 21.5235 | 0.7492 | 31.6691 | 0.8733 | 43.1827 | 0.9767 |
| FedBN | 21.8890 | 0.7495 | 31.6205 | 0.8701 | 42.2720 | 0.9786 |
| FedMRI | 22.2771 | 0.7499 | 32.2095 | 0.8780 | 42.5368 | 0.9788 |
| FedFDD | **22.6056** | **0.7581** | **32.4497** | **0.8821** | 43.0423 | 0.9791 |

## Appendix D. Baseline Method Details

Here we introduce how we adapt the FedMRI (Feng et al., 2023b), HyperNet (Yang et al., 2022) and Fed-PR (Feng et al., 2023a) approaches to our task, as they are not designed for LDCT denoising task and cannot be directly used for the purpose. However, since they are also proposed for low-level image reconstruction task, comparisons against them can further help enhance the evaluation of our proposed method.

**FedMRI: Specificity-Preservation** (Feng et al., 2023b):

We adapted the idea of freezing the decoder (client-specific) and globally sharing the encoder as in the FedMRI approach to our task. Specifically, we built this on the base network architecture, i.e., RED-CNN, which comprises an encoder and decoder. During the

FL training, we froze each client's decoder and enabled the communication of the encoder. Ultimately, each client has its own decoder and a universal encoder.

**PromptFL: Hyperparameter Prompts** (Yang et al., 2022; Feng et al., 2023a):

As discussed in Related Works, HyperFed (Yang et al., 2022) proposed a personalized FL approach using localized hypernetwork of the CT scanning physical properties for the reconstruction of CT projection imaging. However, our task mainly focuses on LDCT denoising, where we cannot access the physical parameters of the imaging, which therefore limits us from using that for the hypernetwork training. On the other hand, FedPR (Feng et al., 2023a) used visual prompts in the null space of global prompt for the FL paradigm, which is not directly feasible in our case.

However, inspired by both the HyperFed and FedPR, we propose to adapt them to our task by introducing the client-specific information as hyperparameters/prompts in the task. Specifically, we proposed to introduce the CLIP representation vector of our client information as prompts to inform the denoising process. The CLIP vector of `"This is an image of {anatomy} low dose CT"` (anatomy can be `chest`, `abdomen` or `head`) is projected into a 96-dim vector, in line with the number of channels of the network bottleneck. Then the bottleneck feature maps are weighed with the softmaxed prompt vector in the channel dimension. The whole operation can be regarded as an attention mechanism prioritizing the specific channels based on the prompts. The denoising network was then trained with the guided information from the prompts.

