# OpenReview forum: "FedFDD: Federated Learning with Frequency Domain Decomposition for Low-Dose CT Denoising"
_MIDL.io/2024/Conference — MIDL 2024 Poster_

### Official Review · Reviewer_u6nd · 2024-02-26

**Confidence:** 5
**Preliminary Rating:** 4

**Summary:**

The author introduce a Federated Learning with Frequency Domain Decomposition (FedFDD) approach, which decomposes images into different frequency components and then updates high-frequency signals in an FL setting while preserving local low-frequency characteristics.

**Strengths:**

The proposed algorithm is evaluated on LDCT datasets of different organs and our experimental results show that FedFDD can surpass state-of-the-art FL methods as well as both localized and centralized models, especially on challenging LDCT denoising cases.

**Weaknesses:**

I noticed that the author has reviewed many related FL works on natural data, e.g., FedBN, MOON, etc.
However, there are many medical FL works that have been published, especially on the low-level task, reconstruction.
Please see the incomplete list here:
Learning Federated Visual Prompt in Null Space for MRI Reconstruction, CVPR 2023.
Specificity-Preserving Federated Learning for MR Image Reconstruction TMI 2022.
I strongly recommend the author to carefully comparsion those works in your paper.

**Detailed Comments:**

see the Weaknesses

**Justification Of The Preliminary Rating:**

I would like to see the comparison with the related works that I have mentioned, not limited to the differences in method design but also the experimental results, because the current version lacks a review and comparison of related works.

**Questions To Address In The Rebuttal:**

see the Weaknesses

---

> ### Author Response · Authors · 2024-03-18
> **Response to Reviewer u6nd**
>
> Q1: **Comparison with other medical FL works:** *“I would like to see the comparison with the related works that I have mentioned, not limited to the differences in method design but also the experimental results…”*
>
> Response:
>
> Thank you for the reviewer’s suggestion. We have now included the comparisons of our FedFDD method with current state-of-the-art medical FL studies, e.g., the FedMRI and FedPR, in the Introduction of the revised manuscript. Despite these well-designed models with advanced optimizing methods (e.g., null space) to control the bias within different clients during communication, our work focuses on the LDCT denoising with various anatomies in an FL setting, and the design of our proposed model is inherently inspired by the denoising nature. This can be exemplified throughout the extensive comparison and ablation study.
>
> In addition,  we have also conducted **two additional experiments** suggested by the core ideas from the papers "Specificity-Preserving Federated Learning for MR Image Reconstruction"  and "Learning Federated Visual Prompt in Null Space for MRI Reconstruction”.
>
> 1. FedMRI. We adapted the idea of freezing the decoder (client-specific) and globally sharing the encoder as in the FedMRI approach to our task. Specifically, we built this on the base network architecture, i.e., RED-CNN, which comprises an encoder and decoder. During the FL training, we froze each client's decoder and enabled the communication of the encoder. Ultimately, each client has its own decoder and a universal encoder.
> 2. PromptFL: As the physical parameters in HyperFed are not accessible in our LDCT denoising task, and the visual prompt in the null space of the global prompt in FedPR is not directly feasible in our case either, we propose to adapt them to our task by introducing the client-specific information as hyperparameters/prompts in the task. Specifically, we proposed to introduce the CLIP representation vector of our client information as prompts to inform the denoising process. The CLIP vector of *"This is an image of {anatomy} low dose CT"* ({anatomy} can be *chest*, *abdomen* or *head*) is projected into a 96-dim vector, in line with the number of channels of the network bottleneck. Then the bottleneck feature maps are weighed with the softmaxed prompt vector in the channel dimension. The whole operation can be regarded as an attention mechanism prioritizing the specific channels based on the prompts. The denoising network was then trained with the guided information from the prompts.
>
> The comparison results are shown in Table 1 in the revised manuscript. We find that the two methods show better performance than the baseline FedAvg on the Chest and Abdomen. This demonstrates the efficacy of these federated learning schemes  (Feng et al.,2023a,b). However, our proposed FedFDD still shows better overall performance than the compared methods as it is specifically designed for the denoising task. A more detailed description of the baseline methods is provided in Appendix D in the revised manuscript.

---

### Official Review · Reviewer_EtbZ · 2024-02-28

**Confidence:** 4
**Preliminary Rating:** 4
**Final Rating:** 4

**Summary:**

This paper proposes Federated Learning with Frequency Domain Decomposition (FedFDD), a novel personalized FL approach for denoising Low-Dose Computed Tomography (LDCT) images, which tackles the challenges of data heterogeneity and privacy concerns. By decomposing images into frequency components and selectively updating high-frequency signals, FedFDD effectively enhances model generalization across various organs. Utilizing an adaptive frequency mask for decomposition, the method outperforms existing FL in LDCT denoising tasks, particularly in complex cases.

**Strengths:**

- The proposed method is well-motivated, clearly showing the specific challenges of LDCT denoising within the federated learning (FL) framework.
- The proposed module, which selectively updates high-frequency components, is particularly novel and effective for this FL denoising task.
- The evaluation is thorough, including a range of general and classical FL baselines applied to this new task.
- The ablation studies are well-conducted, examining various frequencies and split thresholds as critical components of the proposed method.

**Weaknesses:**

- The overall evaluation is thorough; however, I am curious as to why it lacks comparison with other reconstruction based denoising baseline, such as HyperFed [1]. Is there a specific rationale for excluding this type of method from the analysis?

[1] Ziyuan Yang, Wenjun Xia, Zexin Lu, Yingyu Chen, Xiaoxiao Li, and Yi Zhang. Hypernetwork-based Personalized Federated Learning for Multi-Institutional CT Imaging, June 2022.

**Detailed Comments:**

N/A

**Justification Of Final Rating:**

This paper introduces a novel approach to isolating task-specific image components backed by solid analysis and demonstrates its ability to outperform centralized models. The authors have addressed my concerns satisfactorily, and I maintain my initial acceptance score.

**Justification Of The Preliminary Rating:**

The paper merits acceptance due to its strong contributions: it addresses LDCT denoising challenges within federated learning with a well-motivated approach, introduces a novel and effective module for updating high-frequency components, conducts thorough evaluations against a broad set of FL baselines, and provides insightful ablation studies on key methodological components.

**Questions To Address In The Rebuttal:**

Could the authors offer a concise explanation of how the proposed method manages to achieve superior results, even when compared to a centralized setting?

---

> ### Author Response · Authors · 2024-03-18
> **Response to Reviewer EtbZ**
>
> Q1: ***Difference with HyperFed:*** *“...I am curious as to why it lacks comparison with other reconstruction based denoising baseline, such as HyperFed. Is there a specific rationale for excluding this type of method from the analysis?”**
>
> Response:
>
> HyperFed focused on CT reconstruction (projection data to imaging data) tasks guided via a hypernetwork with geometry parameters and dose levels. Our approach leans on low dose CT denoising tasks (noisy imaging data to clean imaging data). The core assumption of HyperFed is the access to the scanning and geometry parameters, which are not necessary for image-level denoising tasks. Therefore, the two methods are not directly comparable.
>
> However, to address the concern of the method comparison against hypernetwork, we add an experiment to incorporate the unique client information, here, the anatomy as attention to guide our denoising task (can refer to Appendix D). Specifically, we incorporate the anatomy information as text embedding to guide the denoising task. We find the proposed FedFDD fits the denoising task better.
> ***
> Q2: **Explanation of the superior performance compared to a centralized setting:***“…how the proposed method manages to achieve superior results, even when compared to a centralized setting?”*
>
> Response:
>
> The superior performance of FedFDD, even when compared to centralized settings, can be attributed to its novel approach to mitigating domain or feature shift issues (caused by different anatomies) inherent in LDCT images, which is however not specifically considered in the centralized setting and other baselines. FedFDD leverages the inherent characteristics of denoising tasks, particularly through the adaptive management of frequency components. Therefore it achieves overall better performance compared to the baselines, particularly in the more challenging case of Chest and Abdomen CT denoising.

---

### Official Review · Reviewer_oGSF · 2024-03-04

**Confidence:** 2
**Preliminary Rating:** 2
**Final Rating:** 3.5

**Summary:**

The paper studies the problem of denoising LDCT images in a privacy preserving federated learning framework. The paper specifically explores the problem of non-IID nature and heterogeneity of data in different participating sites in the medical setting. The key idea of the paper is that the image can be decomposed in low frequency and high frequency parts, and through analysis the authors show that the low frequency components capture the semantics of the image and the high frequency components capture the noise in the LDCT that needs to be removed. Hence, the proposed methods factorized the model in 2 components: local and global, followed by a merge layer. The local component is not shared in the federated framework and only the global component is shared and updated with the gradients corresponding to the high freq component of the images. The experiments show the benefits of the proposed solution over previous state of the art methods and lays the foundation to utilize larger datasets for training the denoising models leading to better results.

**Strengths:**

* The paper presents a novel idea to separate the task specific components of the image from unrelated information and describes the motivation for the idea with backing analysis.
* The results (in the experimental setup presented in the paper) shows the merit of the method and it is impressive to beat separate centralized models for each anatomy.

**Weaknesses:**

* In the experiment section, a very relevant previous work which also proposes a similar approach of having local and global model parameters has not been discussed or used as a baseline even though it is mentioned in the related works.
* While the promise of increasing the dataset available by combining data from different anatomical structures is interesting, the experiment design is not robust enough to back the conclusion. The data is split in different simulated sites by anatomical structure and does not explain if this is a common setting for the problem. I would assume that each site would have data from different anatomical structures and hence it would have been great if the key idea was validated for different simulated splits than just 1.
* The description of the dataset is incomplete. Two papers with different datasets have been referenced without any statistics of the final dataset used in the study.

**Detailed Comments:**

* Please provide more details about the dataset in terms of size and construction.
* It would be good to describe the typical real world setting and how it motivated the simulated split used in the study.

**Justification Of Final Rating:**

The authors addresses some concerns regarding the paper, including different data splits and more baselines and hence I upgrade my rating to reflect the improvements in the paper with the reviewers comments.

**Justification Of The Preliminary Rating:**

The paper presents a novel idea to improve the data available for learning LDCT denoising models and has interesting experiments to show the value of the proposed method. I have some questions regarding the choice of simulated splits (which is a core part of the problem and the proposed method) and comparisons with a similar related work and I am happy to revise the rating once the questions are resolved.

**Questions To Address In The Rebuttal:**

* Results for more splits such that each site has data from multiple anatomies will be helpful to validate the generalizability of the described approach.
* A discussion on how the work is different from the HyperFed paper and how the proposed method compares with that.

---

> ### Author Response · Authors · 2024-03-18
> **Response to Reviewer oGSF**
>
> Q1: **Comparison with similar related work (HyperFed):***“... a very relevant previous work … has not been discussed or used as a baseline even though it is mentioned in the related works” “A discussion on how the work is different from the HyperFed paper and how the proposed method compares with that.”**
>
> Response:
>
> We assume that the reviewer is referring to HyperFed (Yang et al. 2022) as the relevant previous work. HyperFed focused on CT reconstruction (projection data to imaging data) tasks guided via a hypernetwork with geometry parameters and dose levels. Our approach leans on low dose CT denoising tasks (noisy imaging data to clean imaging data). To address the concern of our method’s comparison against hypernetwork, we added an experiment to incorporate the unique client-specific client information to adapt the method to our task (can refer to Appendix D in the revised manuscript). Specifically, we incorporate the anatomy information as text embedding to guide the denoising task. The quantitative comparison results are shown in Table 1 in the revised manuscript. We find the proposed FedFDD fits the denoising task better.
> ***
> Q2: **Clarification on the simulated split and generalization experiments for more splits:***“It would be good to describe the typical real-world setting and how it motivated the simulated split used in the study.” “...The data is split in different simulated sites by anatomical structure…and hence it would have been great if the key idea was validated for different simulated splits than just 1” “Results for more splits such that each client has data from multiple anatomies will be helpful to validate the generalizability of the described approach.”*
>
> Response:
>
> A common assumption in low dose CT is that each client only accesses specific anatomical information, as described in the work of HyperFed (Yang et al. 2022). We argue that in real practice if an institute has LDCT images of more than one anatomical part, it can still be divided into different specialized clients and utilize our federated learning scheme to foster better performance.
>
> Moreover, to further address the reviewer’s concerns and validate the generalizability of our approach, we conducted an experiment where client 1 includes two anatomies (chest, abdomen 1), client 2 includes abdomen (mutually exclusive with client 1) and client 3 includes head. The results demonstrate a comparable conclusion with the initial experiments (PSNR: 22.6056 vs 22.6209; SSIM: 0.7581 vs 0.7586). We think that the experiment demonstrates the robustness and generalization ability of our method. The added results are included in Appendix C in the revised manuscript.
> ***
> Q3: ***More details of the dataset:*** “*Please provide more details about the dataset in terms of size and construction.*” “*The description of the dataset is incomplete. Two papers with different datasets have been referenced without any statistics of the final dataset used in the study.*”
>
> Response:
>
> We have modified the corresponding part and further enriched the dataset-related content in Appendix B of the revised manuscript. The 2016 dataset (McCollough, 2016) is a subset of the LDCT and NDCT datasets (Moen et al. 2021). Usage of the latter needs an additional license and approval. To clarify the manuscript, we removed the misleading reference.

---

> > ### Comment · Reviewer_oGSF · 2024-03-20
> >
> > Thanks for addressing the review comments.
> >
> > For the new data setting in Appendix, can you provide the comparison with baseline values similar to table 1 on page 7 in main article.

---

> > > ### Author Response · Authors · 2024-03-23
> > > **Response to Reviewer oGSF**
> > >
> > > Thank you for your suggestion. We have updated Appendix C Table 2 (also below) with the comparison with representative baseline methods in the new data setting. It is observed that in the new data setting where client 1 contains data with multiple anatomies, our proposed approach FedFDD can still achieve an overall better performance compared to other baseline approaches. This indicates the good generalizability of our method on alternative data splits.
> > > ***
> > > | Method | PSNR↑ (Chest) | SSIM↑ (Chest) | PSNR↑ (Abdomen) | SSIM↑ (Abdomen) | PSNR↑ (Head) | SSIM↑ (Head) |
> > > | --- | --- | --- | --- | --- | --- | --- |
> > > | LDCT | 15.3388 | 0.6857 | 28.3165 | 0.8213 | 43.8707 | 0.9652 |
> > > | Localized | 22.0033 | 0.7445 | 32.1983 | 0.8776 | **44.3414** | 0.9756 |
> > > | Centralized | 22.4649 | 0.7534 | 32.4423 | 0.8809 | 43.6559 |**0.9816** |
> > > | FedAvg | 21.5235 | 0.7492 | 31.6691 | 0.8733 | 43.1827 | 0.9767 |
> > > | FedBN | 21.8890 | 0.7495 | 31.6205 | 0.8701 | 42.2720 | 0.9786 |
> > > | FedMRI | 22.2771 | 0.7499 | 32.2095 | 0.8780 | 42.5368 | 0.9788 |
> > > | **FedFDD** | **22.6056** | **0.7581** | **32.4497**| **0.8821** | 43.0423 | 0.9791 |

---

### Author Response · Authors · 2024-03-18
**Comment to all reviewers summarising the revision**

We thank all the reviewers for their insightful review and constructive feedback on our manuscript. Reviewers have highlighted the importance of the addressed problem (Reviewer EtbZ), commended the novelty and motivation behind our approaches (Reviewer oGSF, EtbZ), and acknowledged the significant improvements over state-of-the-art methods (Reviewer u6nd). As emphasized by Reviewer EtbZ, our work makes a substantial contribution to the problem of LDCT and offers insights for various applications.

In addition to the point-to-point responses to reviewers, we would like to emphasize the clarifications of two main concerns to resolve misunderstandings.

1. Comparison with HyperFed. The presented FedFDD and HyperFed have different application scenarios. FedFDD focuses on denoising low-dose CT images, while HyperFed aims to reconstruct CT images from raw data like Sinograms. The core assumption of HyperFed is access to the scanning and geometry parameters, which are not necessary for image-level denoising tasks. Therefore, the two methods are not directly comparable. In the revised manuscript, we incorporate the concept of employing a client-specific feature vector from HyperFed into our task and include the findings. We observe that FedFDD outperforms this approach.
2. Comparison with other federated learning strategies for low-level tasks. In the revised manuscript, we extend our literature review of current federated learning algorithms. We add the comparison with a state-of-the-art federated learning method proposed for MRI reconstruction, e.g. FedMRI from TMI 2022. We extend our discussion with related works and find that FedFDD shows overall the best results in our task.

We appreciate the insightful feedback from the reviewers, which we've integrated into the updated manuscript. Notable modifications, marked in color, have been made during this rebuttal period:

1. We add a comparison method inspired by HyperFed.
2. We add a comparison method based on specificity-preserving federated learning.
3. We add an experimental setting where a client contains more than one anatomical structure.
4. We extend our related works with more state-of-the-art federated learning approaches in medical imaging.

---

### Comment · Area_Chair_apXp · 2024-03-19
**paper is open for discussions**

Dear Reviewers The authors have submitted their rebuttal addressing the raised questions. The paper remains open for further discussion and engagement.

---

### Meta-Review · Area_Chair_apXp · 2024-04-03

**Recommendation:** Accept (Poster)
**Confidence:** 4

**Metareview:**

Two of the three reviewers support accepting this work as a poster and believe it will generate meaningful discussions. I believe the rebuttal has addressed most of the important comments. The authors are requested to revise the camera-ready version in accordance with the feedback provided in the rebuttal.

---

### Decision · Program_Chairs · 2024-04-05

Accept (Poster)